# Tropical Andes Radar Precipitation Estimates Need High Temporal and Moderate Spatial Resolution

**Mario Guallpa [1,2,3,\*], Johanna Orellana-Alvear [2,4] and Jörg Bendix [4]**

[1] Subgerencia de Gestión Ambiental, Empresa Pública Municipal de Telecomunicaciones, Agua Potable, Alcantarillado y Saneamiento de Cuenca, Benigno Malo 7-78 y Mariscal Sucre, Cuenca 010104, Ecuador
[2] Departamento de Recursos Hídricos y Ciencias Ambientales, Universidad de Cuenca, Av. 12 de Abril, Cuenca 010207, Ecuador; johanna.orellana@ucuenca.edu.ec
[3] Facultad de Ingeniería, Universidad de Cuenca, Av. 12 de Abril, Cuenca 010203, Ecuador
[4] Laboratory for Climatology and Remote Sensing (LCRS), Faculty of Geography, University of Marburg, Marburg 35032,Germany; bendix@mailer.uni-marburg.de
[\*] Correspondence: mario.x.guallpa@gmail.com

**Abstract:** Weather radar networks are an excellent tool for quantitative precipitation estimation (QPE), due to their high resolution in space and time, particularly in remote mountain areas such as the Tropical Andes. Nevertheless, reduction of the temporal and spatial resolution might severely reduce the quality of QPE. Thus, the main objective of this study was to analyze the impact of spatial and temporal resolutions of radar data on the cumulative QPE. For this, data from the world's highest X-band weather radar (4450 m a.s.l.), located in the Andes of Ecuador (Paute River basin), and from a rain gauge network were used. Different time resolutions (1, 5, 10, 15, 20, 30, and 60 min) and spatial resolutions (0.5, 0.25, and 0.1 km) were evaluated. An optical flow method was validated for 11 rainfall events (with different features) and applied to enhance the temporal resolution of radar data to 1-min intervals. The results show that 1-min temporal resolution images are able to capture rain event features in detail. The radar–rain gauge correlation decreases considerably when the time resolution increases (r from 0.69 to 0.31, time resolution from 1 to 60 min). No significant difference was found in the rain total volume (3%) calculated with the three spatial resolution data. A spatial resolution of 0.5 km on radar imagery is suitable to quantify rainfall in the Andes Mountains. This study improves knowledge on rainfall spatial distribution in the Ecuadorian Andes, and it will be the basis for future hydrometeorological studies.

**Keywords:** weather radar; QPE; radar temporal sampling error; rainfall advection; optical flow method; Ecuador Tropical Andes

---

## 1. Introduction

An adequate representation of spatio-temporal rainfall variability is of utmost importance for many meteorological, hydrological, and ecological studies [1,2]. This particularly holds true for the tropical Andes Mountains, which are affected by a high spatio-temporal variability of precipitation but are at the same time characterized by a lack of hydrometeorological monitoring [3,4]. In this regard, rainfall data acquired from different remote sensors, such as weather radar systems, have become more accessible over the last few years [5]. This technology has allowed us to study rainfall with high spatial and temporal resolutions [6], and it can be used in several applications, such as input data in rainfall forecasting studies [7–11], different hydrological models [12,13], early warning systems for floods [14], erosion studies [1,15], research related to atmospheric chemistry [16], and many others. Meanwhile, cost-effective systems derived from ship radar technology are available to establish radar networks with limited funds at hand [17].

The estimation of the rainfall from radar measurements, particularly in the case of non-Doppler or non-polarimetric systems, is affected by different sources of error as discussed in several studies [10,17–19]. This particularly holds true for mountain areas [20–23]. These errors can be grouped into three classes [24,25]. The first class includes ground clutter and beam blocking, which can be removed by adjusting the radar reflectivity maps. The second class is related to the transformation of radar reflectivity to rain rate. The differences between radar and ground rain rate (bias) can be improved using rain gauge data collected over long periods of time. To conduct bias correction of radar data using rain gauge observations, different methodologies have been used successfully in the past, such as radar–rain gauge ratios, as well as Bayesian and geostatistical techniques [23,26,27]. The final class comprises the correction of sampling errors associated with the temporal spacing of radar scans.

Rainfall accumulation maps derived from quantitative precipitation estimation (QPE) with radar data often represent the actual spatial distribution of rainfall incorrectly due to the temporal sampling frequency [28,29]. According to [30,31], the effect of this temporal sampling could be more significant than other error sources combined, particularly when using temporally low-resolution data. Generally, the change rate of rainfall characteristics (e.g., intensity, spatial coverage, rain cell structure) between images increases as the temporal sampling interval increases [32,33]. This is especially critical for highly dynamic convective cells of limited spatial extension, which is the main source of rainfall in the tropics.

In more detail, the influence of specific factors on the assessment of radar rainfall accumulation in comparison to ground data have been found: (i) The sampling frequency has a negative impact depending on the radar spatial resolution. The authors of [28] found a spatial discontinuity in rainfall patterns for hourly accumulation maps by using a sampling frequency of 5 min and a grid resolution of 0.5 km. Similar results were found by [25,31], who also observed an error reduction when using a coarser spatial resolution. Nevertheless, [30] found that for 1 km and 5-min sampling, the error in 10-min rainfall accumulation compared with the high-resolution data (0.2 km and 50 s) is of a magnitude of 20% of the mean rainfall accumulation, increasing to 35% for 2 km and 10-min sampling. Additionally, the temporal sampling interval has greater influence on error than the spatial resolution [25]. For instance, [30] found that for 2 km and 5-min sampling, the error is 50% of the 10-min rainfall accumulation, which is twice that of a 1-min advection interpolation. A solution to reduce the error is to increase the temporal resolution. The authors of [24] increased the sampling frequency to 2 min with 1-km spatial resolution, whereby the mean error was 14% and 8% for scattered and widespread rainfall, respectively. (ii) The storm speed highly influences the cumulative QPE. The authors of [33] determined that the mean error (ME) of the radar–rain gauge correlation decreases with the increase of the storm speed. However, the ME varies throughout the radar range, because the intensity and spatial extent of the precipitation highly influence the error magnitude. Nonetheless, other error metrics increase considerably with storm speed. For instance, the mean absolute error (MAE) between radar and rain gauge ranges from 0.5 to 1 mm h$^{-1}$ with respect to the events' average rain intensity (5 mm h$^{-1}$) [31]. To improve precipitation estimation using radar images, researchers [25,28,34,35] have recommended the use of temporal interpolation methods to add artificial images between the scan times through precipitation advection methods. These methods are proven to reduce the mean absolute error by more than 60%.

Despite the progress in improving cumulative rainfall estimation using radar data, the abovementioned factors, such as radar sample frequency, storm speed, and radar spatial resolution, still challenge the proper evaluation and correction of QPE [28]. Even though the scanning frequency is not a main factor correctly capturing the rain rate at slow storm movements, in particular, the current needed to obtain products with a high spatial resolution (e.g., 0.5 km) increases the difficulty of determining a proper rainfall estimate for fast-tracking storms.

Fast-tracking storms have a strong influence during times of rainfall accumulation of hydrological interest (e.g., 1–3 h) that need more exploration [32]. Furthermore, many hydrological-related studies, such as [5,14,36] have been performed in relatively low altitudes (0–1000 m a.s.l.). As a consequence,

the sampling frequency effect of radar data on the cumulative QPE in higher altitude mountainous areas, such as watersheds located in the Tropical Andes, is still unknown. In this context, the aim of the current study is to analyze the impact of spatial and temporal resolutions of radar data on QPE accumulation for different rainfall features (e.g., rain field speed, rainfall intensity, spatial rainfall coverage, and rain total volume precipitated in the sub-basin). The research was conducted using the highest global X-band radar (CAXX), which is part of the first weather radar network installed in the Tropical Andes of Ecuador [17]. In more detail, in this study, a precipitation advection model based on the optical flow method was validated regarding its applicability in the complex terrain of the Andes. The validated scheme was then used to enhance the original temporal resolution of radar data (5 min) to 1-min temporal resolution. Finally, the effect of using different data recording frequencies (1–60 min) and spatial resolutions (0.5, 0.25, and 0.1 km) to derive rainfall accumulation was assessed by means of several error metrics.

## 2. Materials

### 2.1. Study Area

The study area comprises the middle and upper part of the Paute River basin (Figure 1 and Table 1), which is known as the Cuenca sub-basin. The Paute basin has an area of 6148 km$^2$, which is located in the depression between the east and west of the Andean escarpment of southern Ecuador [37]. It has a complex topography with elevations ranging from 1840 to 4680 m a.s.l. The basin possesses a high ecological and economic value and provides several ecosystem services, such as water supply and regulation, biodiversity conservation, and storage of carbon [38]. About 40% of the basin is covered by Páramo vegetation (3300–4500 m a.s.l.), which plays a key role in water regulation in the region and thus makes this fragile ecosystem highly important for the local population [4,39]. The middle and upper basins (Cuenca sub-basin, 1605.4 km$^2$) are sources of potable water to more than 580,000 inhabitants and indirectly to more than 2000 users through irrigation, agriculture, cattle raising, and other activities [40].

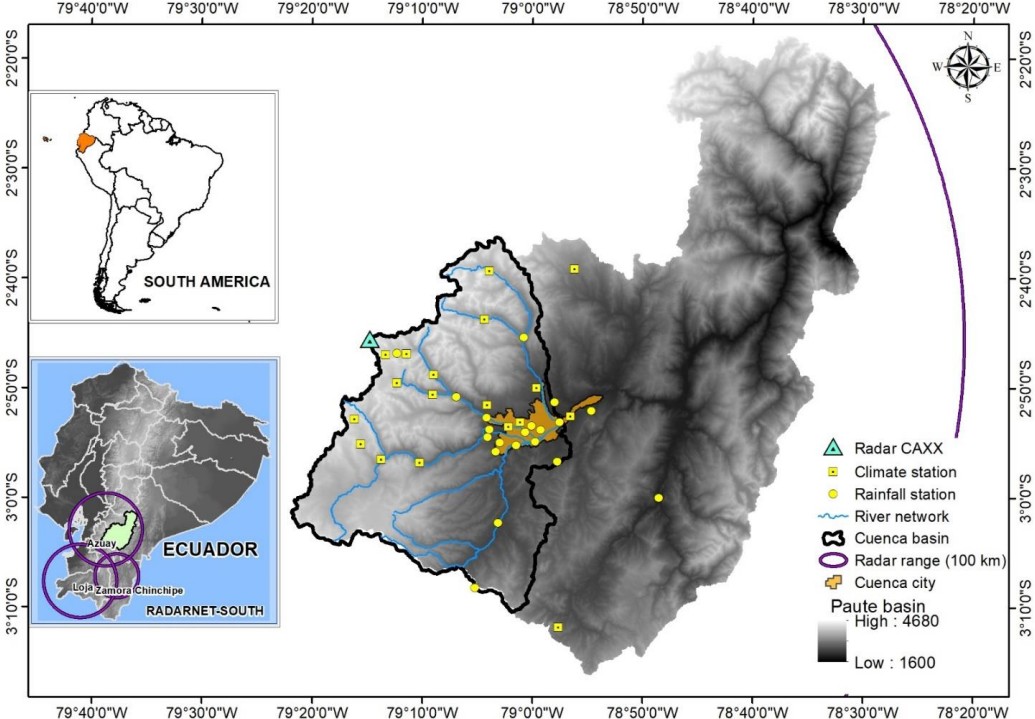

**Figure 1.** Study Area: Paute River basin, CAXX 100-km radar range, rain gauges, and climate stations. Lower left inset: radar network RadarNet-Sur.

The climate in the mountains around the Paute basin is influenced by the Pacific Ocean regime and by the Amazon basin air masses originating from the east [41]. The basin is characterized by a high spatio-temporal precipitation variability [4]. Different rain regimes have been identified based on bimodal inter-Andean seasonality in the basin's central part, with higher rainfall in March–April and October–November and a dry season in July–August [37,42]. The basin's upper part (2600–4680 m a.s.l.) shows bimodal seasonality but with a less pronounced dry season. The average annual rainfall is between 1100 and 1300 mm [4,43,44]. The temperature in the upper part of the basin has a daily average of 8.7 °C ($T_{max}$ = 17.8 °C and $T_{min}$ = 1.8 °C) [45]. The average wind speed at 3900 m a.s.l. is 4 m s$^{-1}$, with maximum peaks of 20 m s$^{-1}$ in the June–September period.

**Table 1.** Characteristics of the middle and upper parts of the Paute River basin.

| Area | Perimeter | Vegetal Cover | Shape | Altitude Min | Altitude Max |
|---|---|---|---|---|---|
| km$^2$ | km | % | - | m a.s.l. | m a.s.l. |
| 1604.5 | 271 | PR (71.3); AMF (10.6) P (7.79); UR (5.7) BQ (2.5) | CO | 2540 | 4680 |
| **Rain** | **Runoff** | **ETo** | **Q. min. daily** | **Q. máx. daily** | **R.C.** |
| mm year$^{-1}$ | mm year$^{-1}$ | mm year$^{-1}$ | mm | mm | - |
| 1221 | 618 | 580 | 0.25 | 1.07 | 0.51 |

CO: Circular to Oval; R.C.: Runoff Coefficient; PR: Páramo; AMF: Andean Montane Forest; QF: Quinoa Forest; P: Pasture; UR: Urban.

## 2.2. Weather Radar Data

The highest weather radar in the world, named CAXX, is part of the new radar network installed in southern Ecuador, RadarNet-Sur [17]. The network consists of three X-band, single-polarized radars. The sensor is a Rainscanner-RS120 Gematronik, located on Paragüillas peak at 4450 m a.s.l. on the northern border of the Cajas National Park near the city of Cuenca (Figure 2).

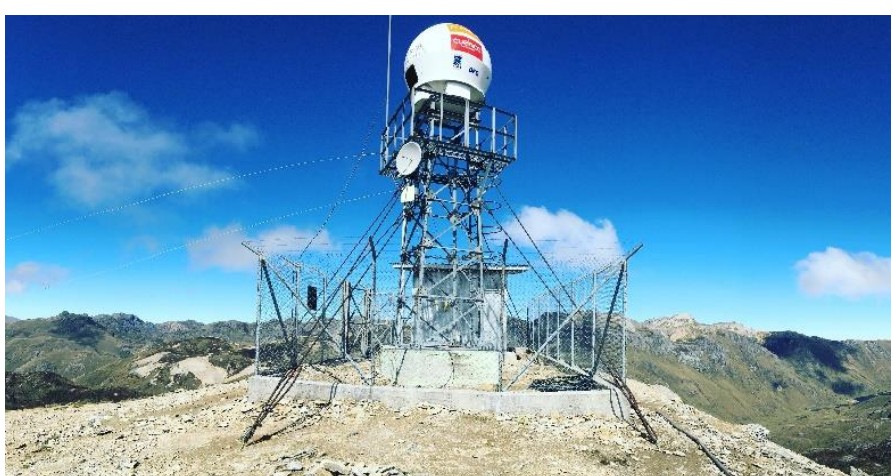

**Figure 2.** X-band radar station: Radar CAXX Rainscanner-RS120 on Paragüillas peak (4450 m a.s.l.).

The CAXX radar covers the Paute River sub-basins and has been operating since April 2015. It has a transmission frequency of 9410 ± 30 MHz, a bandwidth of 2.5 MHz, a pulse length of 1.2–0.5 µs, and a wavelength of 3 cm. The parabolic antenna of the system has a vertical and horizontal beam width of 2°. The radar has a maximum range of 100 km and provides polar images of radar reflectivity every 5 min. The built-in software of the equipment allows for cartesian transformation and produces plan

position indicator images (.ppi) at the same temporal frequency with three different spatial resolutions (0.1, 0.25, and 0.5 km).

The raw reflectivity data needed some corrections to obtain calibrated images, as well as later adjustment using rain gauge data. (i) The built-in software allows for the clutter correction of the raw reflectivities. The removed clutter is the reflection signals produced by solid objects, such as buildings and mountains lying in the radar beam. The resulting blank bins were interpolated with cubic splines (a method incorporated in the RainView Analyzer software) to fill them with values estimated from uncontaminated surrounding reflectivities. According to [1], this method minimizes false estimates within the clutter fields and thus performs better than the bilinear interpolation used by [20]. Then, the corrected reflectivity images were projected to cartesian coordinates with three different spatial resolutions. (ii) To convert reflectivity (Z) to rain rate (R), the site-specific Z–R relationship found by [46] $Z = 204R^{1.57}$ was applied to the radar images. (iii) Afterwards, a bias correction between data of the rain gauge network (G) and the radar (R) CAXX images (RG–RR) was implemented in order to obtain adjusted rainfall products. The difference in the total rainfall between the rain gauge and its corresponding radar pixel was interpolated using the inverse distance weight method. These differences were added only on radar pixels that detected rainfall.

We assumed that bias correction is able to compensate for the radar uncertainty caused by radar calibration and signal attenuation. The radar processing library ωradlib [47,48] was used to obtain the adjusted radar images.

### 2.3. Rain Gauge Data and Rain Events

The rain gauge network is composed of 38 automatic tipping-bucket gauges with 0.254- and 0.1-mm resolution and 5-min time step records. All stations were regularly maintained and operated by ETAPA EP Staff and researchers at the Universidad de Cuenca.

A total of 11 rainfall events were examined from March 2015 to December 2016 (Table 2 and Figure 3). For the rain event selection, three criteria were considered: (i) events with very high rain intensities, (ii) events taking place in the wet or dry season of the study area, with the objective of analyzing the entire rainfall seasonality, and (iii) events with a large spatial extent, covering at least 20 rain gauges within the study area. For each event, the maximum rain intensity was calculated considering a temporal aggregation of 5 min and 1 h. Details for all events are shown in Table 2.

**Table 2.** List of 11 selected rain events that occurred between 2015 and 2016.

| ID Event | Initial Date | Final Date | Duration | Min. Volume Rain Gauge | Max. Volume Rain Gauge | Max. Intensity (5 min) | Max. Intensity (h) |
|---|---|---|---|---|---|---|---|
| | mm/dd/yy h:mm | mm/dd/yy h:mm | h | mm | mm | mm h$^{-1}$ | mm h$^{-1}$ |
| 1 | 04/17/2015 00:00 | 04/17/2015 03:00 | 3:00 | 1.10 | 32.10 | 62.40 | 17.10 |
| 2 | 04/20/2015 16:00 | 04/20/2015 20:00 | 4:00 | 1.27 | 47.00 | 110.40 | 32.20 |
| 3 | 04/21/2015 14:00 | 04/21/2015 18:00 | 4:00 | 1.60 | 27.20 | 40.80 | 17.10 |
| 4 | 05/03/2015 15:00 | 05/04/2015 01:00 | 10:00 | 6.30 | 50.29 | 76.20 | 24.89 |
| 5 | 03/08/2016 14:00 | 03/08/2016 17:00 | 3:00 | 1.50 | 28.96 | 103.20 | 23.00 |
| 6 | 03/10/2016 00:00 | 03/10/2016 06:00 | 6:00 | 1.90 | 30.48 | 34.80 | 13.20 |
| 7 | 04/27/2016 17:00 | 04/27/2016 22:00 | 5:00 | 1.20 | 22.70 | 63.60 | 14.50 |
| 8 | 06/12/2016 12:00 | 06/12/2016 16:00 | 4:00 | 1.20 | 20.07 | 39.62 | 13.46 |
| 9 | 09/14/2016 16:00 | 09/14/2016 22:00 | 6:00 | 2.03 | 38.30 | 74.40 | 29.10 |
| 10 | 10/24/2016 15:00 | 10/24/2016 18:00 | 3:00 | 1.60 | 25.30 | 57.59 | 20.20 |
| 11 | 11/10/2016 17:00 | 11/10/2016 00:00 | 7:00 | 2.90 | 34.30 | 73.20 | 25.30 |

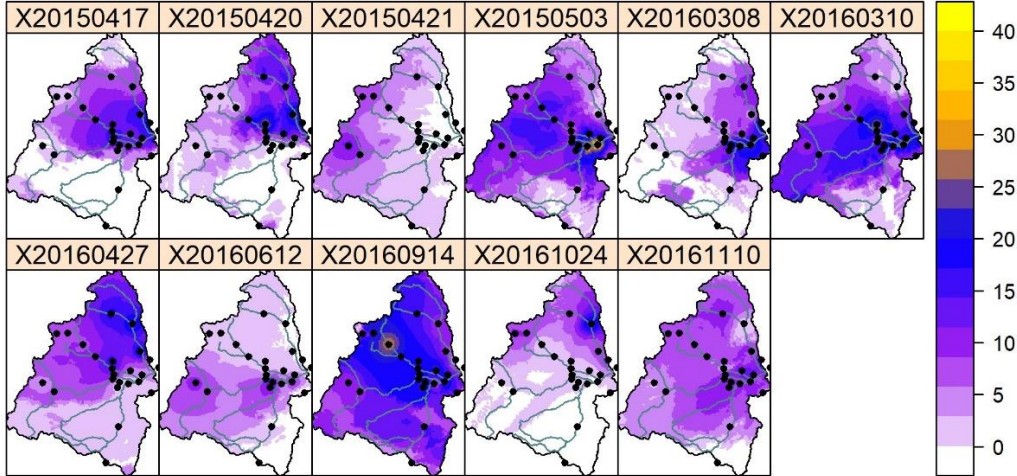

**Figure 3.** Rain distribution and total accumulation (mm per event) for the time period shown in Table 2 (duration) within the Cuenca sub-basin for the 11 events. Black dots represent the rain gauge network. The maps were created using the adjusted radar data.

*2.4. Re-Analysis Data*

Re-analysis data (FNL) of wind speed and direction were used for the comparison with the calculated streamlines of the rain cells. These data were obtained from the National Centers for Environmental Prediction (NCEP) of the National Oceanic and Atmospheric Administration (NOAA). The data have spatial and temporal resolutions of 1° and 6 h, respectively [49], and were interpolated for the location of the radar at 4450 m a.s.l. (CAXX Radar location height). To validate the re-analysis data, wind speed and direction records of spatially distributed climate stations over the study area were used (Figure 1). According to the re-analysis data, winds move from east to west in 91% of all cases (from the Amazon toward the Andes), while in 9% of all cases, the winds originate from the Pacific Ocean (Figure 4).

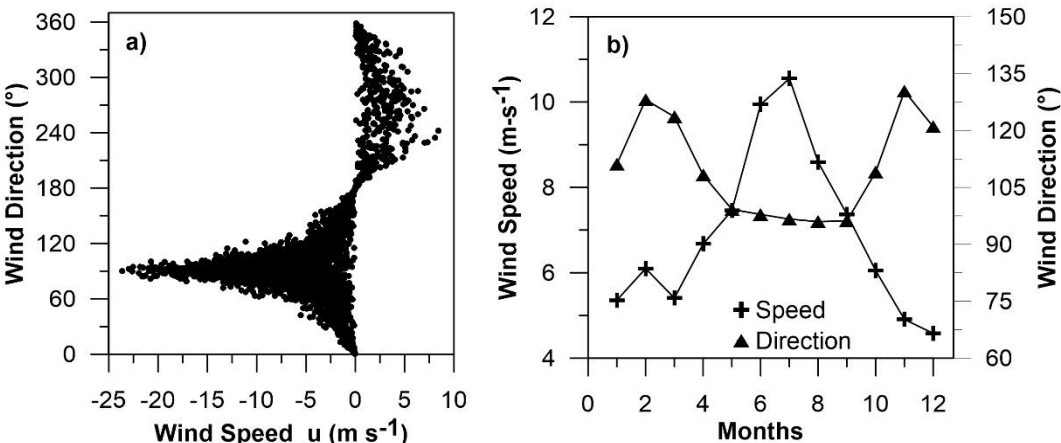

**Figure 4.** Wind speed and direction at 4450 m a.s.l. estimated with the National Centers for Environmental Prediction (NCEP)-FNL data. (**a**) Relationship of the wind speed on the X-axis (positive values towards the east and negative values towards the west) with the wind direction (90° indicates the direction from the east to the west). (**b**) Monthly average distribution of wind speed and direction.

## 3. Methodology

The workflow of the study is depicted in Figure 5. The first step was to enhance the temporal resolution of the radar data. The radar captures rain images every 5 min. However, for this study, 1-min-interval rain images were required. Therefore, rain motion vectors between two consecutive

image slots were calculated through a rain advection model and validated for the local conditions. Then, 1-min-interval images were calculated for each rain event by using the motion vector fields. The second step was the estimation of hourly rain maps in the different spatial and temporal resolutions (0.50, 0.25, and 0.10 km; 1, 5, 10, 15, 20, 30, and 60 min), including the artificial subslot 1-min-interval images. Then, a statistical evaluation of hourly radar rainfall maps was conducted with hourly rain gauge totals to show the general accuracy of QPE. Finally, the impact of different temporal and spatial resolutions of radar data on QPE was analyzed for different rainfall characteristics (horizontal rain field speed, rain total volume, spatial coverage percentage, and maximum rain intensity).

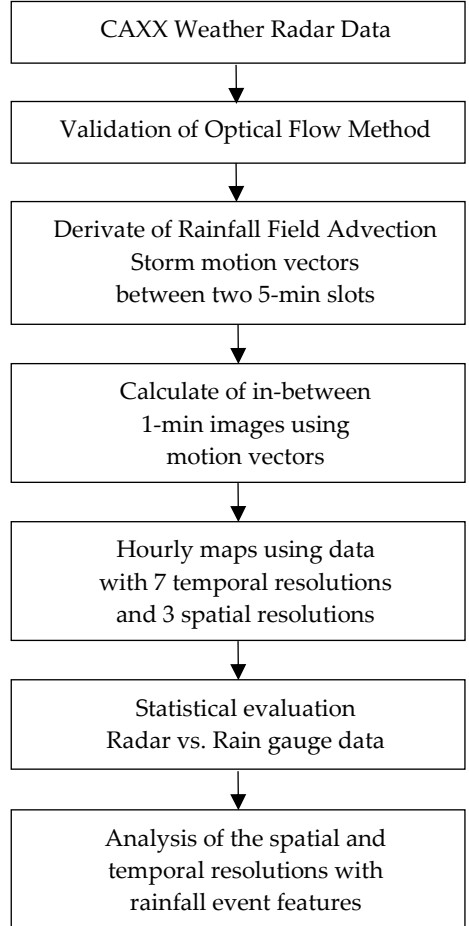

**Figure 5.** Workflow of data processing.

### 3.1. Rain Field Advection

The advection field describes the horizontal motion vectors of rain cells over a certain time period. This movement reflects the rain cells' position change between two consecutive image slots at time t and time t + Δt.

Methods such as the tracking radar echoes by correlation (TREC), which uses the block matching technique [50] and the optical flow using intensity gradient only [51] have already been used to calculate advection. However, these methods showed some inconsistencies in the magnitude of the movement vector [52]. In contrast, the optical flow method using intensity gradient information on several image scales (PyrLK) is a more robust method for an advection model using consecutive images [8,52,53]. The algorithm for this method was introduced by [54] and is based on the Lucas Kanade algorithm [55]. The author of [54] uses an information gradient on different image scales and looks for rain cell speed vectors (magnitude and direction) of two consecutive images. The algorithm has three assumptions: (i) constant radar reflectivities between the pixels of an identified object (rain

cells) of the two consecutive images, (ii) small movements between the image reflectivity cells, and (iii) spatial coherence.

Given two images where F(x) and G(x) are respective reflectivity values at a location x (where x is a two-dimensional (2D) vector), the algorithm looks for a displacement h where G (x) = F (x + h) [54,55]. In the case that h is small (small rain cell motion between two consecutive images), it can be calculated iteratively using Equation (1).

$$h_0 = 0$$

$$h_{k+1} = h_k + \frac{\sum_x w(x)F'(x+h_k)[G(x) - F(x+h_k)]}{\sum_x w(x)F'(x+h_k)^2}$$

(1)

where k is the interaction number until the best h is found by using the Newton–Raphson method. F'(x) and G'(x) are the F(x) and G(x) derivatives calculated by finite differences.

The weight w(x) is defined as

$$w(x) = \frac{1}{|G'(x) - F'(x)|}$$

The method does not calculate the advection vector for each image pixel directly. It only takes the pixels for which the algorithm is most likely to converge. To obtain the entire field information, the calculated motion vector of the selected pixels must be interpolated using the inverse distance weighting method [52]. For the implementation of the optical flow method, the Open Computer Vision (OpenCV) library [56] was used.

To validate PyrLK, two criteria were used: (i) the rain motion vectors were compared with the wind vectors (u, v) determined by re-analysis data using the Pearson correlation coefficient (r), and (ii) by using the rain displacement vectors of each pixel, the radar image was projected for the next 5 min. Then, this predicted radar image was compared with the one observed by the radar at the same time interval. The probability of detection (POD), false alarm (FAR), and frequency of bias (FBI) indices were calculated to quantify the coincidences between the projected and recorded images every 5 min. [8,18,57,58].

Equations (2)–(4) define the formula of these indicators, where tp is the number of true positives (i.e., the pixel number for which the projected and observed images detect precipitation), fp is the number of false positives (i.e., the pixel number for which the projected image detects precipitation while the radar does not), and fn is the number of false negatives (i.e., the pixel number for which the projected image does not detect precipitation while the radar does).

$$POD = \frac{tp}{tp + fn}$$

(2)

$$FAR = \frac{fp}{tp + fp}$$

(3)

$$FBI = \frac{tp + fp}{tp + fn}$$

(4)

Thus, the POD is the precipitation fraction that was correctly detected (POD = 1 is a perfect detection). The FAR measures the precipitation detection fraction that was a false alarm, in other words, the precipitation that was mistakenly detected by the projected image (FAR = 0 indicates perfect detection). The FBI is the radius between the precipitation detected by the projected images in regards to the observations detected by the radar (FBI = 1 indicates perfect detection).

*3.2. Derivation of 1-Min-Interval Images*

For each rainfall event, the derivation of 1-min-interval images between two 5-min-interval consecutive radar images is performed in two steps according to [26]: (i) the first step is to determine

the horizontal rain motion vectors in the X and Y direction for each pixel of the rainfall images (e.g., estimated by the optical flow method explained in Section 3.1). (ii) The motion vectors are used to calculate the artificial 1-min sub-slot images through a linear interpolation of the motion vectors. This procedure was performed for each spatial resolution of the radar images (0.50, 0.25, and 0.1 km). We should point out that the use of a linear interpolation scheme is supported, because rain patterns are preserved between two consecutive images even at high wind speeds [32].

The rain field linear projection at time $t + \Delta t_i$, between time t and $t + \Delta t$, is given by Equation (5), where $R_r(x,y,t)$ denotes the two-dimensional map at time t, x and y represent the coordinates for each pixel, and a represents the displacement between two 5-min-interval consecutive images (b = before, f = forward) [28].

$$R_r(x, y, t + \Delta t_i) = w_1 R_r\big(x + a_{x,f},\ y + a_{y,f}, t\big) + w_2 R_r\big(x + a_{x,b}, y + a_{y,b}, t + \Delta t\big) \tag{5}$$

The displacement a (Equations (6) to (9)) is defined as the number of pixels based on the product resolution (0.1, 0.25, and 0.5 km), and w is a weight factor (Equations (10) and (11)) that is calculated by the distance proportion equivalent to the time contribution within the period interval (5 min).

$$a_{x,f} = \frac{\Delta t_i}{\Delta t} \times \Delta x \tag{6}$$

$$a_{y,f} = \frac{\Delta t_i}{\Delta t} \times \Delta y \tag{7}$$

$$a_{x,b} = -\left(1 - \frac{\Delta t_i}{\Delta t}\right) \times \Delta x \tag{8}$$

$$a_{y,b} = -\left(1 - \frac{\Delta t_i}{\Delta t}\right) \times \Delta y \tag{9}$$

$$w_1 = 1 - \frac{\Delta t_i}{\Delta t} \tag{10}$$

$$w_2 = \frac{\Delta t_i}{\Delta t} \tag{11}$$

### 3.3. Impact of Spatial and Temporal Resolutions of the Radar Data on Cumulative QPE

The hourly-accumulated rainfall observations (G) at rain gauge locations are compared with the corresponding radar pixel values (R) from the hourly accumulation maps. For the evaluation, the Pearson correlation coefficient (r), the bias index (BIAS), and the root-mean-square error (RMSE) were determined between G and R [18,58]. In addition, the POD, FAR, and FBI indices were calculated to quantify the coincidences between data detected by radar and by rain gauges.

### 3.4. Rainfall Features

Four rainfall features were analyzed: rainfall field motion, rain total volume, spatial coverage percentage, and maximum intensity. The rainfall events were split into three ranges according to the average motion of the rainfall events (1–2.5, 2.5–5, and >5 m s$^{-1}$). Then, we analyzed whether the radar–rain gauge relations show similar behavior under these three speed ranges. The POD and BIAS indexes were used to analyze the R–G relation for the three spatial and seven temporal resolutions.

To the second and third rainfall feature, the rain total volume (m$^3$) and spatial coverage percentage were calculated for each event over the Cuenca sub-basin (1604.5 km$^2$). The two features were calculated using the entire spatio-temporal configuration of the radar data. Next, with the aim to illustrate the results of the radar data, the same steps for total rain and coverage were estimated using the rain gauge network data only. Despite their use in the recent literature, Kriging methods could not be applied in this study for two reasons: (i) the available rain data lack a normal distribution and (ii) the rain data could not be adjusted to any theoretical semivariograms. Therefore, in spite of

their limitations, the Thiessen and inverse distance weight (IDW) methods were used to generate the interpolated rain maps.

Finally, the maximum rainfall intensities (mm h$^{-1}$) of the events were analyzed. The analysis and discussion focused on the influence that maximum rainfall intensities have on the speed of the rain fields, as well as the total volume and the spatial coverage of the rainfall.

## 4. Results

### 4.1. Validation of Optical Flow Method

The average speed of most of the events did not exceed 6 m s$^{-1}$ (Figure 6), which is in agreement with the wind records measured by climate stations and reported by [44,45] for the study area. Most of the rain fields moved from east to west according to the predominant wind direction.

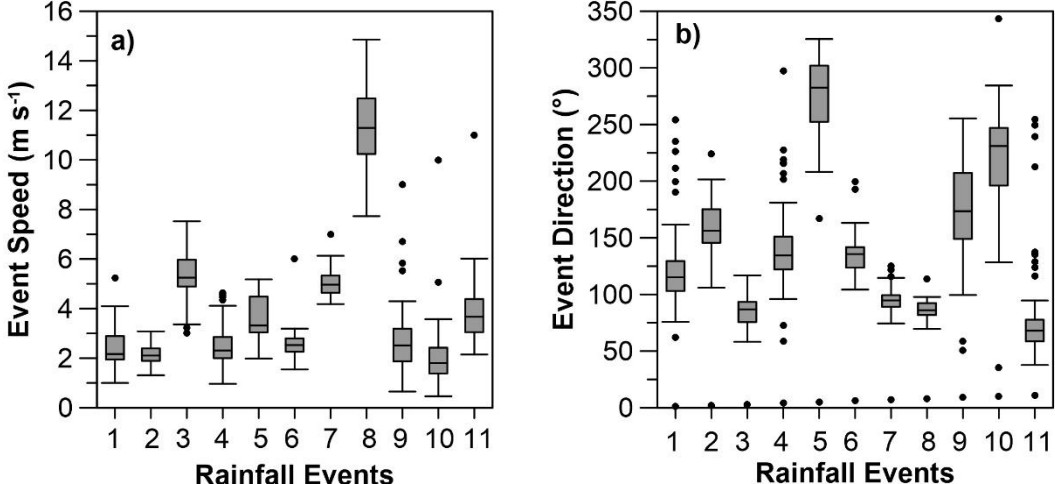

**Figure 6.** Distribution of (**a**) the speed and (**b**) the direction of the 11 rain events, calculated with the optical flow method using intensity gradient information on several image scales (PyrLK) advection model. Event direction means the angle of the origin of the rain cell (e.g., 90° = origin from the east).

Figure 7 shows that there is a strong correlation between the events' average motion and the wind speed. Although the correlation coefficient r was high at 0.91, the wind speeds were greater than the advection velocity of the rain cells. For instance, the maximum wind speed for event 8 was 19 m s$^{-1}$, while the average speed of the event reached 11.25 m s$^{-1}$. The other events showed the same behavior. This is because the precipitating clouds usually move at a lower speed than the wind that pushes them [37,59].

The PyrLK model performance was evaluated by comparing the observed and estimated radar images created by the model during the same period (Table 3). According to the advection model, the rainfall event with the maximum average speed of movement (11.25 m s$^{-1}$) occurred in June 2016. This speed agrees with the austral summer period and the usual presence of strong winds for the study area in this season [37,44] (Figure 4). For this event, r and the POD for the projected and observed images were the lowest, 0.62 and 0.82, respectively. In contrast, for event 10, which had the lowest speed (1.92 m s$^{-1}$), r and the POD (0.79 and 0.86) were above the average of these indices (0.75 and 0.85). This indicates that the advection model is more accurate for events with slow motion of rain fields. However, for similar intermediate speeds (2.5 m s$^{-1}$), lower correlation coefficients (0.68–0.75) were found. This is because PyrLK is best adjusted when the rain event characteristics meet the model assumptions between consecutive images (brightness/constant intensity and small movements in the rain cell shapes) [28,54].

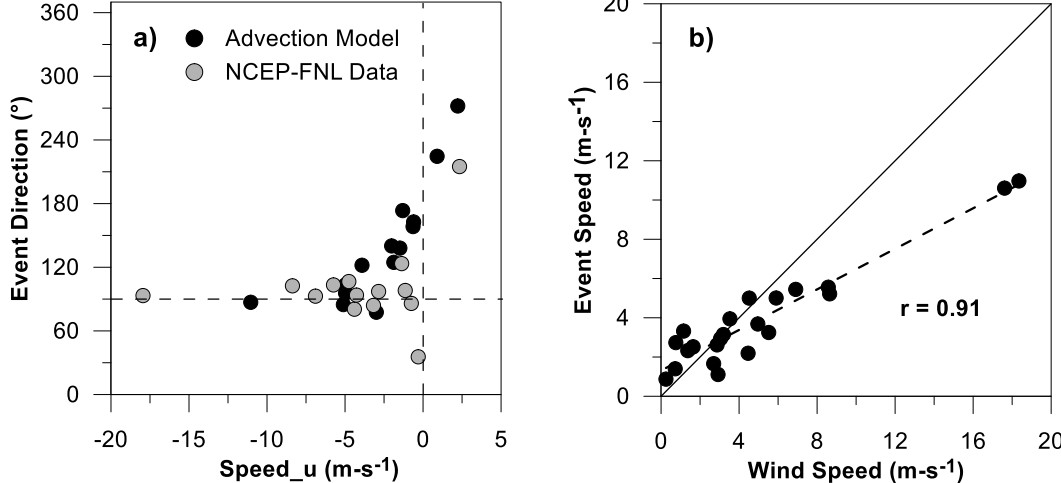

**Figure 7.** (**a**) Distribution of the wind speed and direction of the re-analysis data in contrast to the rainfall motion generated by the optical flow method for all precipitation events (negative event speed indicates that precipitation is moving west). (**b**) Hourly speed correlation generated by re-analysis data (X-axis) and the advection model (Y-axis) for the 11 events.

**Table 3.** Correlation coefficient r and average detection index for each rain event between images estimated by the advection model and observed by radar every 5 min.

| ID Event | r * | Detection Index | | |
|---|---|---|---|---|
| | | POD | FAR | FBI |
| 1 | 0.83 | 0.90 | 0.19 | 1.11 |
| 2 | 0.70 | 0.82 | 0.35 | 1.38 |
| 3 | 0.70 | 0.84 | 0.29 | 1.19 |
| 4 | 0.74 | 0.88 | 0.27 | 1.24 |
| 5 | 0.68 | 0.78 | 0.42 | 1.45 |
| 6 | 0.78 | 0.86 | 0.26 | 1.20 |
| 7 | 0.75 | 0.87 | 0.40 | 1.61 |
| 8 | 0.62 | 0.82 | 0.31 | 1.22 |
| 9 | 0.79 | 0.91 | 0.18 | 1.14 |
| 10 | 0.79 | 0.86 | 0.20 | 1.10 |
| 11 | 0.75 | 0.89 | 0.25 | 1.25 |

* The *p*-values of r were lower than 0.05 for the 11 rainfall events.

As expected, the model detected fewer false alarms in events with slower motions (events 1, 9, and 10) as shown by the FAR. On the other hand, the PyrLK model generally overestimated the rainfall detection; this can be observed in the FBI indicators for all the events. This tendency was more marked for events with fast motions (events 7 and 8). Nonetheless, the average r and POD for the 11 events was 0.75 and 0.85, respectively, which, according to [58,60], statistically allows the PyrLK model to be appropriate for the study site, and consequently, the model is suited for use in the atmospheric conditions of the Tropical Andes. In addition, when using the optical flow method to estimate the rain cell motion vectors, even storms with speeds greater than 12 m s$^{-1}$ can be captured by consecutive images [28].

### 4.2. Impact of Spatial and Temporal Resolution of the Radar Data on QPE

Table 4 summarizes the different statistical and detection indexes between hourly accumulated data for the radar–rain gauge relation of the 11 events. It can be seen that r slightly improved (from 0.67 to 0.69) when using 1-min-interval data instead of 5-min-interval data for the three spatial resolutions. The authors of [28] found similar results in their study, which analyzed the image resolution effect on the radar–rain gauge relation.

The BIAS decreased 0.1 mm on average for the three spatial resolutions when 1-min-interval data was used. This is equivalent to enhancing the estimate by 40% by using 1-min-interval data instead of 5-min-interval data, which seems to be consistent with the results (30%–60%) found by [31]. Moreover, in agreement with the previous results, the detection index improved significantly (10%) when using 1-min-interval interpolated images.

On the other hand, when the different indexes from Table 4 are compared for the three spatial resolutions, it was found that the differences are negligible. There is no considerable improvement (<5%) in the radar–rain gauge relation if the spatial resolution increases from 0.50 to 0.10 km. Consequently, 0.50-km pixel images can be used for radar–rain gauge analysis in the study area, which will result in lower computational costs for information processing.

**Table 4.** Statistical error metrics and detection indexes between radar data and rain gauges for different spatial and temporal resolutions of radar images.

| Sample Time (min) | 0.50 km | | | | | |
| --- | --- | --- | --- | --- | --- | --- |
| | r * | BIAS (mm) | RMSE (mm) | POD | FAR | FBI |
| 1 | 0.69 | −0.15 | 2.46 | 0.76 | 0.13 | 0.98 |
| 5 | 0.68 | −0.21 | 2.46 | 0.71 | 0.15 | 0.89 |
| 10 | 0.65 | −0.28 | 2.42 | 0.68 | 0.16 | 0.84 |
| 15 | 0.6 | −0.34 | 2.45 | 0.66 | 0.16 | 0.78 |
| 20 | 0.53 | −0.4 | 2.5 | 0.62 | 0.19 | 0.73 |
| 30 | 0.44 | −0.47 | 2.54 | 0.56 | 0.2 | 0.66 |
| 60 | 0.34 | −0.72 | 2.85 | 0.42 | 0.23 | 0.48 |
| | 0.25 km | | | | | |
| 1 | 0.67 | −0.11 | 2.49 | 0.79 | 0.12 | 1.03 |
| 5 | 0.67 | −0.23 | 2.46 | 0.71 | 0.15 | 0.87 |
| 10 | 0.65 | −0.29 | 2.42 | 0.68 | 0.15 | 0.83 |
| 15 | 0.56 | −0.36 | 2.45 | 0.65 | 0.16 | 0.78 |
| 20 | 0.5 | −0.39 | 2.49 | 0.62 | 0.18 | 0.72 |
| 30 | 0.4 | −0.47 | 2.53 | 0.56 | 0.19 | 0.65 |
| 60 | 0.3 | −0.75 | 2.89 | 0.41 | 0.24 | 0.47 |
| | 0.10 km | | | | | |
| 1 | 0.68 | −0.13 | 2.48 | 0.8 | 0.12 | 1.08 |
| 5 | 0.67 | −0.26 | 2.46 | 0.71 | 0.15 | 0.86 |
| 10 | 0.65 | −0.3 | 2.41 | 0.68 | 0.15 | 0.83 |
| 15 | 0.58 | −0.36 | 2.45 | 0.65 | 0.16 | 0.77 |
| 20 | 0.51 | −0.4 | 2.48 | 0.61 | 0.18 | 0.72 |
| 30 | 0.4 | −0.49 | 2.57 | 0.55 | 0.18 | 0.64 |
| 60 | 0.31 | −0.74 | 2.84 | 0.41 | 0.24 | 0.47 |

* The *p*-values for r were lower than 0.05.

The statistical and detection indexes improve as the radar data temporal resolution increases (Table 4 and Figure 8). The correlation coefficient decreased from 0.69 to 0.31 (45%) when the sampling time increased from 1 to 60 min. The POD showed similar behavior (decrease of 49%). The BIAS increased by 0.35 mm (73%) when the radar temporal resolution was reduced to 30 min. Similar results (20%–60%) were found by [19] with 30-min-interval radar data. The FBI index indicated that the radar data underestimation increased to 52% when the radar recorded images every 60 min in comparison to 1-min sampling, and the FAR detected by the radar increased to 50% with images recorded every 60 min.



Similar results were observed with all three pixel resolutions. However, a comparison of radar-based QPE pixels and gauge-based point observations always implies sources of uncertainty, which prevent perfect validation results: (i) The radar–rain gauge relationship worsens with lower temporal resolution, because radar images only show an integral area of the pixel, but the local (subpixel) rainfall might be higher over the gauge than integrated over the entire radar pixel [25,30,31]. Furthermore, the rain gauge network is unable to record the high spatio-temporal precipitation variability of the Tropical Andes and might miss pixel-integrated rainfall if the subpixel location of the pixel remains rain-free [4,44].

(ii) Another source of uncertainty is the horizontal displacement of falling rain observed by the radar through wind drift on its way to the rain gauge. The vertical wind profile, between the radar signal and the surface, modifies the raindrops' precipitation trajectory [61]. If the low temporal resolution of the radar is added to this effect, the deviation between QPE and the rain gauge total increases [62–64]. The altitude differences between the radar beam (around 4450 m a.s.l.) and the rain gauge network are between 550 and 2000 m. The wind profile effect on the QPE–rain gauge relation has not yet been studied for the Tropical Andes' atmospheric conditions. For this, the height and depth of the melting layer needs to be known (for instance using a vertical profile radar). Current studies using a micro rain radar (MRR) close to the study area are under development and could be used in the future.

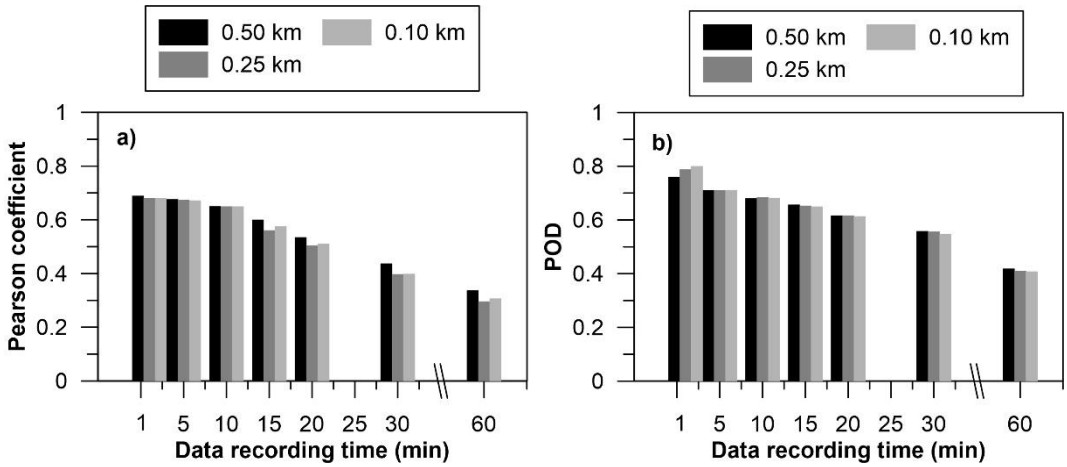

**Figure 8.** Variation of the Pearson correlation coefficient (**a**) and the probability of detection index (**b**) according to the sampling time and pixel resolution.

Despite the fact that results recorded in up to 10-min-intervals are likely the most relevant, an investigation of higher integration times (10–60 min) was performed because of the possible importance for remote areas of the Tropical Andes. In this area, most rainfall monitoring networks just use manual automatic weather stations (i.e., collect rainfall data three times per day only). Only a few institutions have installed a limited number of automatic weather stations (AWS) so far. Thus, this study offers these institutions evidence of the limitations of temporally low-resolution data collection and provides insights into future AWS network planning regarding the frequency needed to properly capture the rainfall variability.

### 4.3. Influence of Rain Event Features

Table 5 shows the three groups of rainfall events based on rainfall motion speed.

**Table 5.** Precipitation event groups according to rainfall motion.

| Rain Motion Range (m s⁻¹) | Date | | Rain Motion (m s⁻¹) | | |
|---|---|---|---|---|---|
| | **Start** | **End** | **Vx** | **Vy** | **V** |
| 1–2.5 | 10/24/2016 15:00 | 10/24/2016 18:00 | 0.90 | −0.64 | 1.92 |
| | 04/20/2015 16:00 | 04/20/2015 20:00 | −0.64 | −1.77 | 2.41 |
| | 04/17/2015 00:00 | 04/17/2015 03:00 | −1.88 | −0.59 | 2.43 |
| | 03/10/2016 00:00 | 03/10/2016 06:00 | −1.48 | −1.51 | 2.45 |
| | 05/03/2015 15:00 | 05/04/2015 01:00 | −2.01 | 0.36 | 2.46 |
| 2.5–5 | 09/14/2016 16:00 | 09/14/2016 22:00 | −1.30 | −0.70 | 2.64 |
| | 03/08/2016 14:00 | 03/08/2016 17:00 | 2.22 | 1.45 | 3.54 |
| | 11/10/2016 17:00 | 11/10/2016 00:00 | −2.99 | 1.71 | 3.71 |
| >5 | 04/27/2016 17:00 | 04/27/2016 22:00 | −4.97 | −0.12 | 5.18 |
| | 04/21/2015 14:00 | 04/21/2015 18:00 | −5.11 | 0.66 | 5.42 |
| | 06/12/2016 12:00 | 06/12/2016 16:00 | −11.04 | 0.97 | 11.24 |

### 4.3.1. Rain Field Motion

Figure 9 shows the POD and BIAS discretization of the radar–rain gauge relation as a function of the average rainfall motion (RM). There is a strong relationship between the POD and BIAS with the RM. The POD values were higher in the event of higher RM (RM > 5 m s⁻¹). The lowest POD values occurred for RM between 1–2.5 m s⁻¹. Although the POD differences for 1-min-interval recorded images were less than 6% between the three speed ranges, this percentage increased to 11% after 5 min. However, the difference again decreased for 60 min (5%).

The BIAS has a similar behavior to the POD regarding the RM (Figure 9). The errors were greater for lower RM (1–2.5 m s⁻¹) and decreased for the other two ranges. The BIAS differences for events between 2.5 and 5 m s⁻¹ and >5 m s⁻¹ were low (11%) compared with the lower RM range (41%). These results are in agreement with those obtained by [31], who mentioned that the error in the radar–rain gauge relation decreases with the increase in the storm speed. In the same way, [28] found lower r and higher RMSE for events with lower RM.

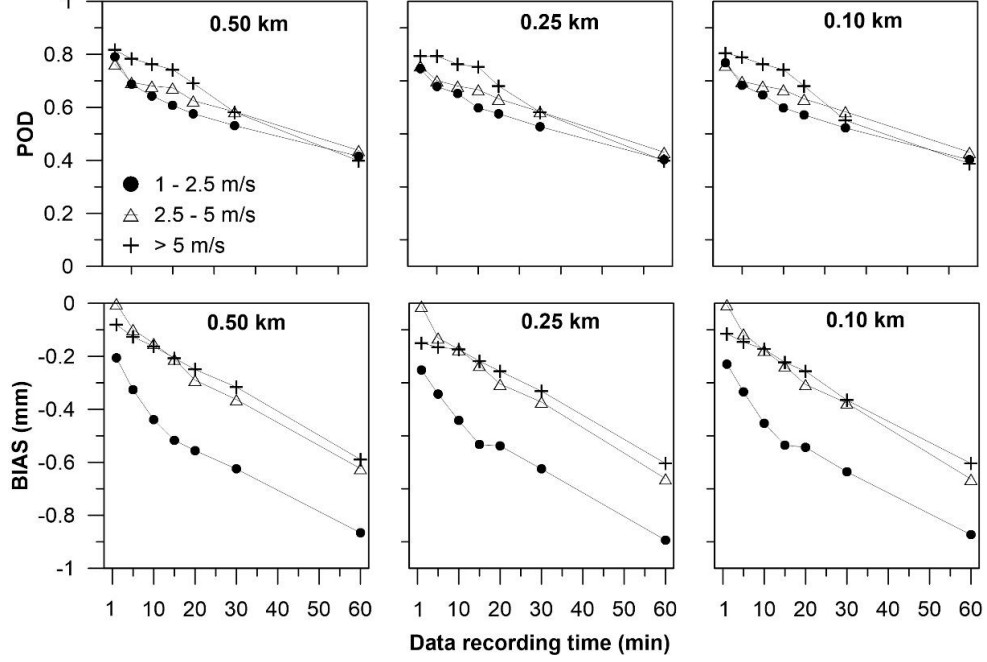

**Figure 9.** Probability of detection and BIAS indexes for precipitation events according to average rainfall motion. Subplots for different temporal and spatial resolutions of radar data are shown.

4.3.2. Rain Total Volume and Spatial Coverage Percentage

Figure 10 shows a clear tendency of decreasing rain volumes when the radar sampling time increases. On average for all events, the rain total volume improved by 7% when using 1-min-interval images in comparison with 5-min-interval images and by 12% in comparison with 10-min-interval images. On the other hand, we calculated that the error of rain total volume using the three pixel resolutions was less than 3%. This result confirms that radar images with a 0.5-km resolution can be used for hydrological studies in this region.

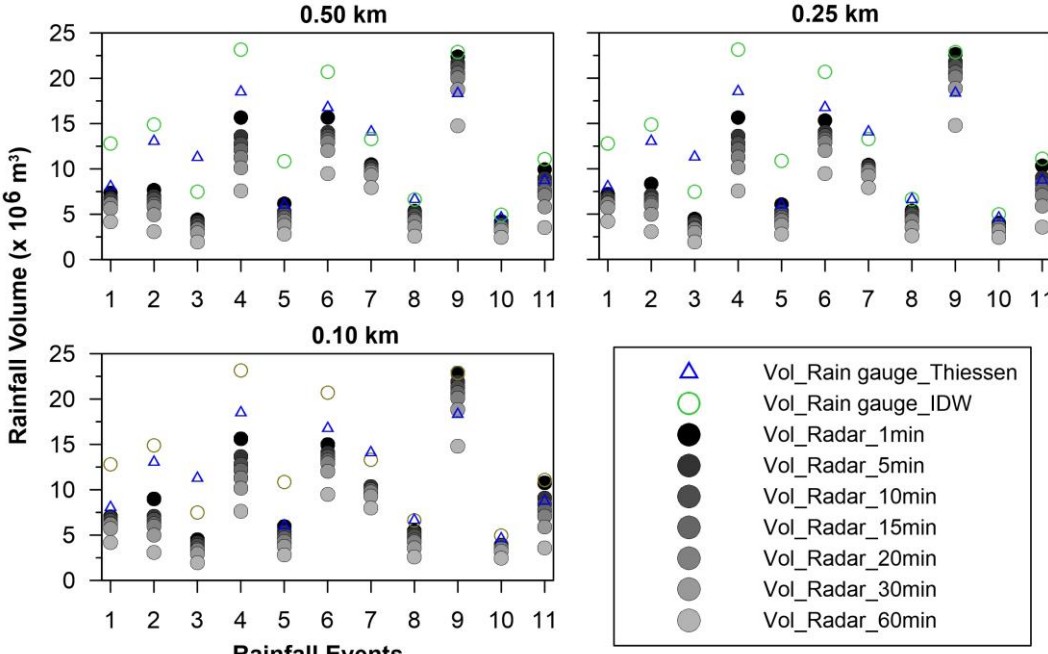

**Figure 10.** Accumulated rainfall volume (m³) for the 11 events according to the different radar temporal and spatial resolutions.

There is a remarkable relationship between the spatial rainfall coverage over the basin and the rain total volume (Figure 11). This relationship is not linear, because it depends on other event characteristics, such as intensity and duration [28]. In this case an exponential relation was found (r = 0.79). Nonetheless, the coverage percentages estimated with the rain gauge network differed from those estimated by the radar data. For instance, for the event on 04/17/2015, using only the rain gauge network data, the rainfall precipitated over 80% of the sub-basin's total area calculated with the Thiessen method and 100% of the sub-basin's total area calculated with the IDW method (Figure 12). Nonetheless, when using the higher temporal resolution, 1-min-interval radar data, with 0.5-km pixel size, it was identified that the event covered only 58% of the basin. A possible explanation for this might be that the IDW method interpolates the rain for the whole study area, even where no rainfall is recorded. For this reason, the estimated rain volumes are greater than those calculated using the Thiessen method and those observed by the CAXX radar.

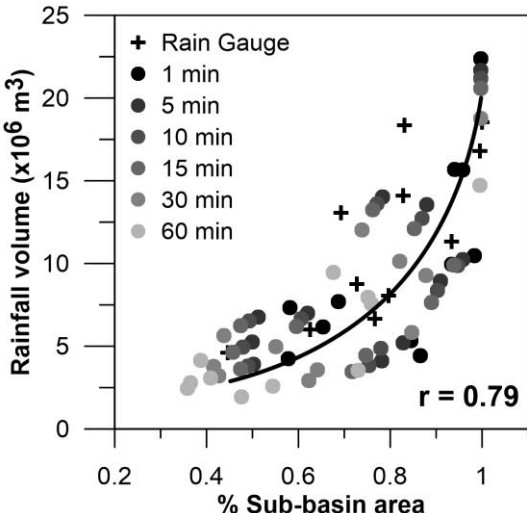

**Figure 11.** Relation of water volume and its percentage of coverage over the study basin area.

The coverage–rain volume relation significantly influences the accumulated rainfall estimation during a certain period. In this case, as in most cases of Tropical Andes studies where rain gauges are used to estimate rainfall [37,57,65,66], there is a sparse rain–gauge network, especially in the sub-basin's headwaters. The insufficient monitoring added to the high precipitation variability present in the study area [1] can cause two scenarios: (i) rainfall underestimation when a part of the precipitation event is not captured by the rain gauge network and (ii) rainfall overestimation when the tributary area for each rain gauge is considerably high (e.g., the Thiessen method).

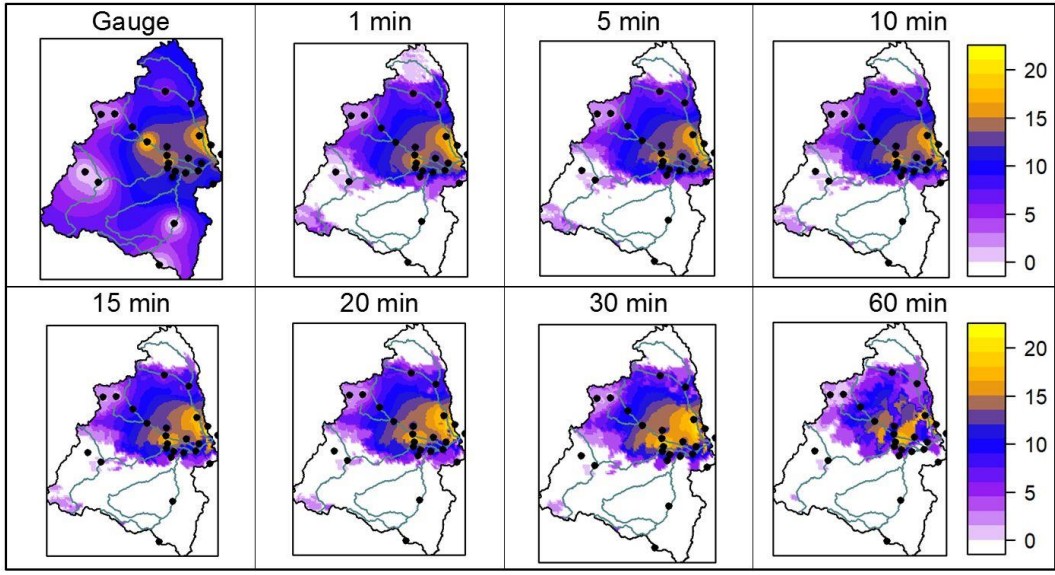

**Figure 12.** Accumulated rainfall (mm) for the event on 04/17/2015. Rain spatial distribution for the basin using only rain gauge network data (upper left) and radar data for different sampling times with 0.5-km spatial resolution.

### 4.3.3. Maximum Rainfall Intensities

Figure 13 shows the relation among the maximum rainfall intensities (MRI), the rainfall motion (RM), rainfall volume, and spatial coverage percentage of the 11 events. Events with an RM greater than 5 m s$^{-1}$ presented relatively low MRI for hourly data (13.46–17.10 mm h$^{-1}$). The highest MRIs estimated with 5-min-interval data were recorded for the events between the RM ranges of 1–2.5 and 2.5–5 m s$^{-1}$, with values of 110.3 and 103.20 mm h$^{-1}$, respectively. These results can be attributed to the

different types of precipitation that are generated in these ecosystems [44,46]. The heavy rain structure consists mainly of drops with mean volume diameters (Dm) greater than 1 mm [46], which in turn makes it difficult for the wind to move them easily, therefore obtaining a low average rainfall motion ($1–2.5\ \text{m s}^{-1}$).

In contrast, moderate ($0.5 < \text{Dm (mm)} \le 1.0$) to light rains ($0.1 < \text{Dm (mm)} \le 0.5$) can be pushed relatively easily by the wind, thus generating rain motion speeds greater than $5\ \text{m s}^{-1}$. Finally, when light rain occurs in months with higher wind speeds (e.g., the event on 06/12/2016), the RM can reach values up to $15\ \text{m s}^{-1}$ (Figure 6, event 8).

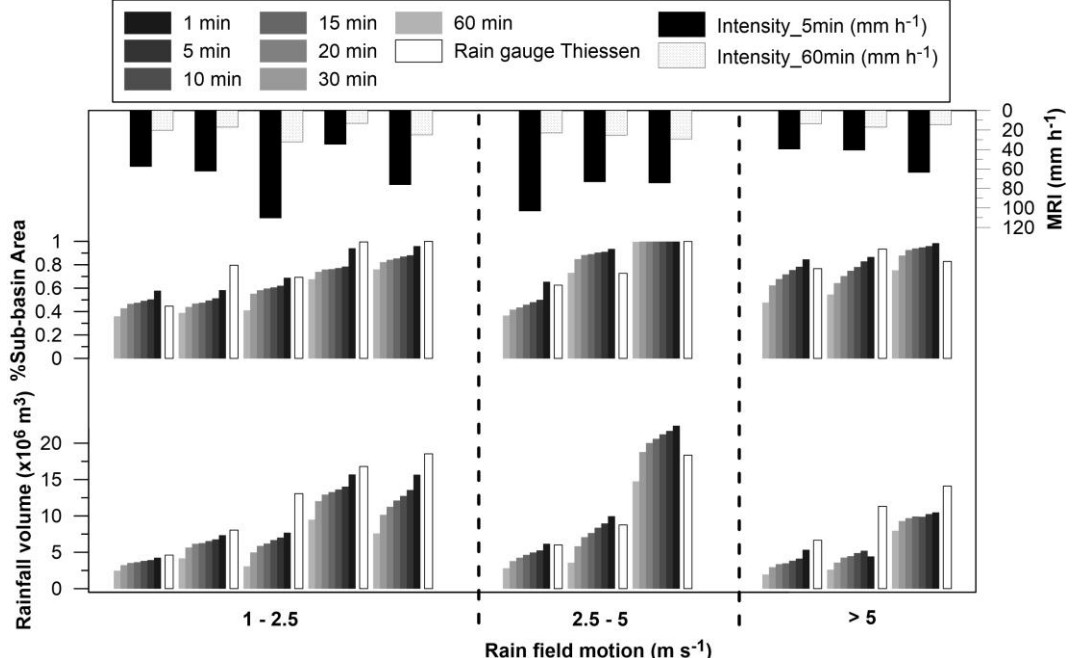

**Figure 13.** Event discretization according to the horizontal rainfall motion for the seven temporal resolutions and for 0.5-km spatial resolution (the same results were found when using the two other spatial resolutions). The precipitated water volume, the area percentage covered by rainfall, and the maximum intensities recorded for 5 min and 60 min are shown.

When the RM ($2.5–5\ \text{m s}^{-1}$) of event 3 (9/14/2016) of group 2 was analyzed (Figure 13), it was identified that the rainfall precipitated over the entire basin as calculated by the radar data and rain gauge method. The rainfall accumulated by the radar was higher than that estimated by the rain gauge network. This can be explained by the fact that the rain cells with heavy intensity were not captured by the rain gauges. In contrast, in event 1 of the same group (03/08/2016), where the second highest intensity was recorded ($103.20\ \text{mm h}^{-1}$), the rain gauge network properly captured the cell with the heaviest intensity.

On the other hand, the event with the highest MRI ($110.3\ \text{mm h}^{-1}$, event 3 of the RM range $1–2.5\ \text{m s}^{-1}$, Figure 13) registered on 04/20/2015, presented a similar basin coverage percentage for the rain gauge network as well as the radar (using 1-min-interval data). However, the rain gauge network registered 42% more precipitation than the radar. A possible explanation for this might be that the greatest amount of rainfall was precipitated under the radar beam. The CAXX radar is installed at 4450 m a.s.l., the average height of the basin is 3454 m a.s.l., and Cuenca city, where some heavy rain events also occur, is located at 2560 m a.s.l. This suggests that the radar beam does not record rainfall events below a certain range.

## 5. Conclusions

In this study, the impact of spatial and temporal resolutions on the QPE accumulation under different rainfall features was analyzed. The optical flow method (PyrLK) was validated and used afterwards to determine the field vectors of the rainfall motion. Rainfall field advection was applied to enhance the radar data temporal resolution to 1 min through a liner interpolation between two 5-min-interval consecutive images. We used three different spatial resolutions and seven temporal resolutions to assess the sampling time impact on QPE under four rainfall features: horizontal rainfall motion, spatial precipitation coverage, rain total volume precipitated into the sub-basin, and rainfall intensity. The following conclusions were obtained from the study:

(a)　The PyrLK model can be used in the atmospheric conditions in the Tropical Andes. The average correlation coefficient and probability of detection index between the images recorded by the radar and estimated by the model for the 11 analyzed events were 0.75 and 0.85, respectively. However, it was observed that the PyrLK is better adjusted for events with slow rain motions (r = 0.79, POD = 0.86) than for fast motions (r = 0.62, POD = 0.82). Thus, the slower the horizontal rain motion, the higher the model performance.

(b)　The radar–rain gauge relation was enhanced when using 1-min-interval radar data instead of 5-min-interval data (r from 0.67 to 0.69) for the three pixel resolutions. The BIAS between the radar and rain gauge decreased by 40% and the POD significantly improved (11%) when using 1-min-interval interpolated images.

(c)　There was a slight improvement (<5%) in the radar–rain gauge relation when the pixel resolution is increased from 0.50 to 0.10 km. Therefore, for radar–rain gauge analysis in the study area, images of 0.50 km of spatial resolution are a good trade-off, which will reduce the computational cost of data processing.

(d)　The hourly radar–rain gauge relation decreased considerably with the increase of the data sampling time (r from 0.69 to 0.31 for recording times of 1 to 60 min). These tendencies may be explained by the fact that by decreasing the temporal resolution, the radar does not capture the rapidly evolving echoes or rain cells. In addition, the rain gauge network does not capture the high spatial variability of the Tropical Andes precipitation.

(e)　In temporal resolution terms, using 1-min-interval radar data enhanced the rain total volume estimation by 7% compared with 5-min-interval radar data and 12% compared with 10-min-interval radar data. This improvement occurred, because the 1-min-interval radar images were able to capture more details of the rain event characteristics.

(f)　The difference between the spatial rainfall coverage captured by the radar and the rain gauge network significantly influenced the calculated rain volume. This difference occurred mainly because (i) the rain gauge network is neither dense nor homogeneous, and it does not capture all of the rain cells and may (strongly) underestimate or overestimate the rainfall between stations through interpolation processes; and (ii) some precipitations may be generated below the height of the beam radar. Efforts are currently being carried out to fill this knowledge gap by using a combination of the CAXX horizontal radar and a vertically-pointed MRR to determine the vertical profile of rain and identify the height at which the precipitation events are generated.

The study gives important insights into the proper development of hydro-meteorological networks in the high Andes, which is urgently needed due to the great importance of ecosystem ecohydrological services. On the other hand, the results lead to an improvement of rainfall estimations in mountain areas in order to better determine the entire water balance in the region. In addition, the results will form the basis for implementing a precipitation nowcasting model, which, in turn, will help to enhance the rainfield flow forecasting in mountainous areas, e.g., in rainfall hazard surveillance systems.

**Author Contributions:** M.G. Conceptualization, Methodology, Software, Data curation, Formal Analysis, Writing—Original draft. J.O.-A. Conceptualization, Software, Writing—Review & Editing, Supervision. J.B. Writing—Review & Editing, Visualization.

**Funding:** The current study was developed in the context of three projects. (i) The knowledge transfer project "RadarNet-Sur" (http://www.radarnetsur.gob.ec/), administered by the University of Marburg and the Provincial Government of Loja (GPL) within the "Platform for Biodiversity and Ecosystem Monitoring and Research in South Ecuador". This was funded by the German Research Foundation (Deutsche Forschungsgemeinschaft—DFG; BE1780/31-1 and BE1780/38-1). (ii) The project "Identificación de los procesos hidrometeorológicos que desencadenan crecidas a partir de la información suministrada por un radar de precipitación" and (iii) the project "Desarrollo de modelos para pronóstico hidrológico a partir de datos de radar meteorológico en cuencas de montaña " These were funded by the Research Office of the University of Cuenca (DIUC) and Empresa Pública Municipal de Telecomunicaciones, Agua Potable, Alcantarillado y Saneamiento de Cuenca (ETAPA-EP).

**Acknowledgments:** The authors thank Rolando Célleri for helping to greatly improve the manuscript and Ing. José Medina for practical support of the computational implementation.

**Conflicts of Interest:** The authors declare no conflicts of interest. The funders had no role in the design of the study; in the collection, analyses, or interpretation of data; in the writing of the manuscript, or in the decision to publish the results.

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
