# Peer review of "Tropical Andes Radar Precipitation Estimates Need High Temporal and Moderate Spatial Resolution"

_water, doi:10.3390/w11051038_

Round 1

Reviewer 1 Report

The paper is generally well written and conveys the idea sufficiently. This study enriches scientific knowledge, and it can publish as a present form.

Author Response

We thank the reviewer for her/his comment. It encourages us to follow our research on radar rainfall in mountain regions as the Andes. 

Reviewer 2 Report

I think that this is a very interesting manuscript.

I suggest to Authors to enrich the introduction by citing other approaches for combining radar and rain gauge data, for example via Bayesian techniques:

Todini, E.: A Bayesian technique for conditioning radar precipitation estimates to rain-gauge measurements, Hydrol. Earth Syst. Sci., 5, 187-199, https://doi.org/10.5194/hess-5-187-2001, 2001

Seck, I., Van Baelen, J. Geostatistical merging of a single-polarized X-band Weather Radar and a sparse rain gauge network over an urban catchment (2018) Atmosphere, 9 (12), art. no. 496, .Kim, T.-J., Kwon, H.-H., Lima, C. A Bayesian partial pooling approach to mean field bias correction of weather radar rainfall estimates: Application to Osungsan weather radar in South Korea (2018) Journal of Hydrology, 565, pp. 14-26.Oriani, F., Ohana-Levi, N., Marra, F., Straubhaar, J., Mariethoz, G., Renard, P., Karnieli, A., Morin, E. Simulating Small-Scale Rainfall Fields Conditioned by Weather State and Elevation: A Data-Driven Approach Based on Rainfall Radar Images (2017) Water Resources Research, 53 (10), pp. 8512-8532.

Moreover, Authors could introduce the importance of rainfall lattice data for nowcasting of space-time rainfall fields:

Xu, H., Wu, Z., Luo, L., He, H. Verification of high-resolution medium-range precipitation forecasts from global environmental multiscale model over China during 2009-2013 (2018) Atmosphere, 9 (3), art. no. 104, .

Versace, P., Sirangelo, B., De Luca, D.L. A space-time generator for rainfall nowcasting: The PRAISEST model (2009) Hydrology and Earth System Sciences, 13 (4), pp. 441-452.

Author Response

We thank the reviewer for his/her suggestions. We have added these reference articles in the introduction.  Please see new lines 41 and 51 - 53.    

Reviewer 3 Report

I recommend to make the manuscript much more concise

Author Response

We thank the reviewer for his/her suggestions. We have revised the manuscript accordingly, methodology and results sections have been modified and clarified. Specifically paragraph 1 of general methodology; paragraph 1 of section 3.2; paragraphs 1-3 of section 3.4; paragraphs 5-7 of results section 4.2.

We have splitted the result section 4.3 into three subheadings to make it more understandable. Finally the conclusions are according to methodology and results.  

Reviewer 4 Report

The topic of the manuscript is very interesting and original because it studies data from the world’s highest located radar at 4450 m a.s.l. The manuscript deals with using radar data for precipitation estimation.
In general, the scientific content of the article needs only several modification, but the presentation requires improvement. A lot of sentences are inaccurate and sounds strange. Some of them are mentioned below but not all.
May general comment on Section 4.3 and differences between areas with “radar” and “gauge” precipitation is that the authors try to solve problem that cannot be solved. You cannot obtain similar areas if you have several discrete measurements which you interpolate by IDX method. I recommend you to remove or at least shorten this part of the paper. 
Figure captions should contain description of all variables including units.

Specific comments:
Title: I think that “deriving optimal settings” is not good wording. This gives the impression that the article deals with hardware radar setup.
In my opinion you should concentrate only on 1, 5 and 10 min. time resolution because existing meaning is that larger intervals are useless. Especially comparisons of the results for 1 and 60 min. and showing that 60 min. data are worse is not fair because it is clear and in advance known result.
Line 145: What do you mean by “adjustment images”?
Line 148: Could you give some additional information about the interpolation method? Cubic splines may cause oscillations and even negative values in the interpolated field. How do these problems are solved or do they not occur?
Line 152: I would recommend using “to measured reflectivity”.
Line 163: “that records …” – please reformulate.
Line 166-170: Please reformulate the criteria. E.g. “events taking place all over the year …” cannot be fulfilled.
Line 224: Please reformulate “of the precipitation event”.
Line 235: You almost do not use term “reflectivity”, which is what a radar measure. Here, for example, you speak about brightness in other parts of the text you speak about imagines. Please try to use term reflectivity as it is common.
Line 235: I do not know what you mean by “constant brightness between the same pixels of the two consecutive images“. It is true only for two identical fields.
Line 271-277: Please reformulate this paragraph. I think that you do not say what you want to say.
Line 293: I do not understand this sentence (collocated …).
Line 298: What do you mean by “for 1-hour comparison”? Do you compare accumulated precipitation?
Line 302: I do not agree with your arguments that kriging cannot be applied. On the contrary, the argument a) is an argument why IDX should not be used.
Line 315 and below: I am surprised by low speed 6 m/s, which is not the case in Europe. Also argumentation by speed at climatic stations cannot be applied in Europe. Are you sure with your statements? I think it would help if you describe the studied events from the viewpoint of convective or stratiform precipitation. You mentioned that you selected cases with large amount of precipitation then I suppose that convection was dominant.
Line 327-328: Could you add a reference?
Line 342: Instead of “higher performance” I recommend “more accurate”.
Line 377: “statistical coefficients” are not appropriate words.
Line 381-387: Please be accurate in your statements. For example, you say that “The statistical and detection indexes decrease as the radar data recording time increases”. Is it true for FAR?
Line 392 and below: Could you add same simple calculations about possible movement of rain drops due to horizontal wind using estimated terminal velocity?
Line 411: I do not understand this sentence.
Line 414: I do not understand this sentence.
Line 426: I do not understand what you want to say by “properly” in this context.
Line 433: I do not understand (whether).

Author Response

We thank the reviewer for her/his the positive feedback, comments and observations. We have prepared a document with the replies to solve your questions. Please find this file attached. 

Round 2

Reviewer 3 Report

Personally, I feel the manuscript has not yet reached a scientific paper Standard. It is certainly an interesting technical Report, but I think it is not yet adequate for publication. I would recommend reject and re-submit in a much more concise form with better illustrated and more focused Targets.